# Relative Validity of an Italian EPIC Food Frequency Questionnaire for Dietary Factors in Children and Adolescents. A Rizzoli Orthopedic Institute Study

**DOI:** 10.3390/nu13041245

**Published:** 2021-04-09

**Authors:** Inge Huybrechts, Rossella Miglio, Lorenza Mistura, Sara Grioni, Irene Pozzebon, Carmen Odorifero, Renato Borea, Angelo Gitto, Mariela Terrafino, Mattia Scipioni, Aida Turrini, Vittorio Krogh, Susanna Ricci, Gianfranco Martucci, Alessandra Longhi

**Affiliations:** 1International Agency for Research on Cancer, World Health Organization, 69372 Lyon, France; huybrechtsi@iarc.fr; 2Dipartimento Scienze Statistiche “Paolo Fortunati”, University of Bologna, 40126 Bologna, Italy; rossella.miglio@unibo.it; 3Council for Agricultural Research and Economics, Research on Food Nutrition, 00178 Roma, Italy; lorenza.mistura@crea.gov.it (L.M.); aida.turrini@crea.gov.it (A.T.); 4Fondazione IRCCS Istituto Nazionale dei Tumori, Epidemiology and Prevention Unit, 20133 Milano, Italy; sara.grioni@istitutotumori.mi.it (S.G.); vittorio.krogh@istitutotumori.mi.it (V.K.); 5LILT, Lega Italiana Lotta Contro Tumori Association, 00178 Roma, Italy; irene.pozzebon01@gmail.com (I.P.); Carmen.Odorifero@legatumori.mi.it (C.O.); mattiascip@gmail.com (M.S.); gfmartucci@gmail.com (G.M.); 6Epidemiology and Prevention Unit (EPIC), University Federico II, 80135 Napoli, Italy; renatomik2002@hotmail.it; 7Istituto Ortopedico Rizzoli, IRCCS, Dietetic Service, 40136 Bologna, Italy; gittoangelo.ag@gmail.com (A.G.); mterrafino2000@yahoo.it (M.T.); susanna.ricci@ior.it (S.R.); 8Istituto Ortopedico Rizzoli, IRCCS, Chemotherapy Unit, 40136 Bologna, Italy

**Keywords:** FFQ, 3-day diary, validity, food intake, bone sarcoma, child, adolescent

## Abstract

Dietary factors play a major role in the development of non-communicable diseases, however little is known regarding the impact of nutrition on rare diseases like sarcomas. This Rizzoli Orthopedic Institute study aimed to evaluate the relative validity of a Food Frequency Questionnaire (FFQ) to measure the consumption of foods in comparison with a 3-days diary diet in a healthy Italian student population aged between 12 and 17 years. An extended version (including food groups for children) of the semi-quantitative FFQ used in the European Prospective Investigation into Cancer and Nutrition (EPIC) was administered. The validity of the FFQ was assessed by comparing the intakes from the FFQ against the 3-day diary method. 254 Italian subjects were included in the analyses: 128 females; 126 males; 116 from High Secondary School (14–17 years); 138 from Low Secondary School (12–13 years). Mean and median intakes are overall higher in the FFQs than in the food diaries. Spearman correlations adjusted for within-person variability were highest for legumes, vegetables and coffee/tea (>0.5), followed by potatoes, meat, fruits, breakfast cereals, biscuits and candies, and milk/yoghurts (>0.4). Moderate correlations were found for alcoholic drinks, soft drinks, juices, and grains (>0.3). For some food groups, such as fish, potatoes, and bread, correlations tend to become higher when stratifying the analyses for age group. These results demonstrate that the adapted EPIC COS FFQ validated in Italian adults is also appropriate and well understood by Italian children and adolescents.

## 1. Introduction

Several dietary intake assessment methods, i.e., 24-h dietary recalls, dietary records, and food frequency questionnaires (FFQ), can be used to assess the diet of individuals. FFQs are often used in epidemiological studies to estimate long-term dietary exposures, mainly due to applicability in large samples and their ability to categorize subjects based on their intake [1,2,3]. The FFQ that was used in the European Prospective Investigation into Cancer and Nutrition (EPIC) study is one of the most extended questionnaires for food consumption, but it has been developed for adults only. Food frequency questionnaires (FFQ) for adults are not necessarily applicable to children due to different comprehension skills and food consumption habits. A FFQ records the intake of a limited list of carefully selected food items, and is as such developed for a specific purpose and dietary risk factors. In addition, as food intake and dietary habits vary by population groups, FFQs are often developed and used for specific populations. Therefore, it is important to consider also the specific diet of children and adolescents when assessing dietary risk factors in these younger age groups. To date, the validity of a comprehensive FFQ to assess dietary risk factors has not been evaluated in Italian children and adolescents. The existing studies, albeit with some conflicting results, have been able to highlight important associations between diet and cancer risk using an FFQ. However, these instruments either did not provide validity estimates, or were validated for use in adults only. Measurement errors in FFQs can be estimated through relative validation studies, in which dietary intake, assessed via the FFQ, is compared with a more precise method, such as dietary records [1]. In this study an adapted version of the Italian EPIC FFQ (EPIC COS) [4], which includes all dietary factors related to cancer, was evaluated for validity in children and adolescents between 12 and 17 years of age. The EPIC COS FFQ was used as a basis because it was considered as the most recent dietary questionnaire used in an adult Italian population of younger age (<40 years old) to investigate interactions of genetic predisposition and diet and environmental risk factors for the detection of early breast cancer in women <40 years old [4]. Therefore, the purpose of the present study was to develop and evaluate the validity of a FFQ among children and adolescents in Italy to be used in a case control study in bone sarcoma patients older than 12 years and adults (Metabol Sarc NCT04734289).

## 2. Materials and Methods

### 2.1. Study Population

To validate the adapted EPIC COS FFQ among children and adolescents, we conducted a validation study in 14 different classes of which 8 low secondary schools (scuola secondaria di primo grado, from 12 to 13 years of age) and 6 high secondary schools (scuola secondaria di secondo grado, from 14 to 17 years of age) between 1 April 2019 and 30 November 2019. Nine different schools were included in the analyses, with 5 schools in Northern Italy, 1 school in Central Italy and 3 schools in Southern Italy. The distribution of the classes over the different regions included 3 low and 3 high grade classes in the north, 2 low grade classes in central Italy and 3 low grade and 3 high grade classes in the south. The distribution per city included 1 low grade school in Milano and 1 high grade school in Novi Ligure (Alessandria), 2 high grade schools in Bologna, 1 low grade school in Padova, 1 low grade school in Roma, 1 high grade schools in Termoli (Campobasso), and 1 low and 1 high grade school in Napoli.

The FFQ was administered to 280 child/adolescent students from 12 years old up to 17 years old. Food diaries recorded on three non-consecutive days were used to validate the questionnaire in healthy students from the North, Center and South of Italy. Considering the inherent challenges to dietary intake assessment studies using burdensome open-ended methods such as food diaries, we considered a potential dropout (or exclusion of incomplete diaries) of 10% when deciding upon the number of subjects to include in our validation study. Therefore, a total number of 280 subjects were recruited to obtain at least 250 students with high quality food diaries, which is a suggested number needed to validate a FFQ [5].

Considering the Italian population with its multi-cultural and regional eating habits, a stratification in the three main Italian macro areas (North, Center and South) was used for the sampling of the schools to be included in the study. The distribution of residents in the three considered macro areas was: 28 Million in Northern, 12 Million in Central, and 20 Million in Southern Italy respectively in 2019 (http://dati-censimentipermanenti.istat.it/?lang=it# accessed on 5 April 2021) and we enrolled 120 students in North, 60 in Center and 100 in the South of Italy. The Italian School system is divided in two school types for the considered age group, namely the low-grade secondary school of 3 years course (11–13 age range years old) and the high grade secondary school of 5 years (13–17 age range). The validation consisted in comparing the FFQ with the 3-D diary record for assessing food intake.

### 2.2. The EPIC COS Food Frequency Questionnaire (FFQ) Adapted for Children

The EPIC COS FFQ is a 163 items self-administered semi-quantitative food frequency questionnaire. An extended version (including food groups for children) of the original semi-quantitative EPIC COS FFQ (which was developed for adults), was developed for children by using more informal language comprehensible for children. It was integrated with few questions about anthropometric measures (weight and height at registration and at birth), the use of a special diet (e.g., vegetarian), parents’ education and residency. The FFQ asked participants to recall the consumption frequency for the past year. Participants estimated the frequency of intakes of each food with different possible frequency options: times per week, times per month, and times per year. The FFQ also assessed the average quantity consumed per day. Photos with examples of different portion sizes (low, medium, large) of different recipes (soup, pasta, steak etc.,) were provided as in the original EPIC COS FFQ to assist participants when estimating the average quantity of food consumed per day. An overview of the different food groups is included in the Annex.

The food items were linked, using specific software [6], to Italian Food Composition Tables [7] to obtain estimates of daily intakes of energy and 37 macro and micronutrients.

The EPIC COS questionnaires were completed by the students in the classroom where an explanation of the food items was provided by the interviewer. One day before this FFQ completion, the interviewer obtained the consent from the parents and the assent from the students. After compilation the interviewer checked any missing or unclear answers and clarified with the student one by one.

### 2.3. Three-Day Food Diaries (Reference Method)

After one week from the administration of the FFQ a new visit was arranged in the classroom in which a description was given about the three (non-consecutive) days CREA (Consiglio per la Ricerca in agricoltura e l’analisi dell’economia agraria) food diary [8] which was administrated by the trained interviewer in each classroom.

### 2.4. Cascade Training for the Collection of the Food Diaries

A one-day course was organized by the CREA-AN (Research Centre for Food and Nutrition) to train six volunteers of the LILT (Lega Italiana Lotta Contro i Tumori) on how to collect food consumption data. In turn, the trained volunteers explained to the students in each school selected where the data collection was carried out, how to report the consumed food for 3 non-consecutive days on the food diaries.

### 2.5. Data Collection of the Food Diaries

The first day the food diary was completed in the classroom for the child’s breakfast and snack of the same day. At home the parts of the other meals and snacks of this first day 1 were further filled out, and the interviewer then checked it the day after with the student one by one for unclear answers. The diary for the second and the third day were compiled in the following days (one of these two days should be a holidays/weekend).

Food diaries were filled out by the students trained according to the following criteria:-For each eating occasion, time, place of consumption, detailed description of foods and beverages, quantity consumed, and brand (for manufactured foods) were carefully checked and recorded in the food diaries.-Portion sizes were assessed with the help of a picture booklet from CREA, developed for assessing food portions (small, medium, large) when completing the diaries. For each of the three days, students had to declare if they followed a particular diet and if the consumption reported was different from their usual one.

Once collected, the volunteers checked all the diaries, then evaluated the registration adding a comment on the reliability of the information recorded on each single day.

### 2.6. Data Recording of the Food Diaries

The trained volunteers inputted the diaries using the FOODCONS software (www.foodcons.eu, accessed on 5 April 2021) that includes four databases: Food nomenclatures, household measurement units, standard recipes, and food composition databases. Once the food item has been inputted it is converted into the weight of single raw ingredients and the corresponding amounts of nutrients intake.

For each inputted food diary the FOODCONS delivers an excel file with the daily intake of total energy, water, as well as of macro and micro-nutrients (protein, fat, saturated or polyunsaturated fatty acid, available carbohydrate, sugars, starch, fibers, alcohol, 6 minerals, and 12 vitamins) at food item level.

The CREA-AN has developed the software FOODCONS and all connected instruments (hard-copy food diary, picture atlas) were tested and adapted for this study: a more informal language addressed to children was used.

### 2.7. Data Analysis

Estimates of dietary intake from the FFQs and 3-day diary, as well as the differences estimated by the two methods, were verified for normality. As for some of the foods the distributions deviated significantly from a normal distribution, in addition to the mean value, median, and inter-quartile range (IQR) were used to describe the estimates.

Spearman correlation statistics are provided to enable comparison with previous studies. These correlation coefficients were also corrected for attenuation due to random error in the reference measurements (within person variability) as described by Liu et al. Variance ratios of within- over between-individual variance were calculated and expressed as *S_w_*^2^/*S_b_*^2^ with *S_w_* the within-person variance and *S_b_* the between-person variance [9].

All statistical analyses were run in SPSS. Only significant correlations were included in the tables (*p* < 0.05 level).

The study protocol and instruments were approved by the Rizzoli ethical committee n.553/2018/Oss/IOR. The aim of the study was clearly explained to the students and their parents and written consent from parents was obtained before the interview.

## 3. Results

A total of 254 students participated in this validation study, including 128 girls and 126 boys. Analyses were run for the overall group of 254 students as well as stratified for age, including 138 students from Italian low secondary schools (12–13 years) versus 116 students of high secondary schools (14–17 years).

As presented in Table 1, mean and median intakes are overall higher in the FFQs than in the food diaries. Except for potatoes, rice, vegetable oils and butter intake, other food groups were overestimated by the FFQ compared to the diary. Differences in mean and median intakes between the two dietary intake assessment methods were smaller when stratifying the analyses for age and gender (See Appendix A
Table A2).

Spearman correlations have been calculated and are presented in Table 2. Correlations were highest for legumes, vegetables, and coffee and tea (≥0.5 for most age groups), followed by potatoes, meat, fruits, breakfast cereals, and biscuits and candies (>0.4). Moderate correlations were found for milk and yogurts, alcoholic drinks, soft drinks, juices, and grains (including pizza) (>0.3). For some food groups, such as fish, potatoes, and bread, these moderate correlations tend to become higher when stratifying the analyses for age group. Correlations for boys and girls are very similar for most major food groups.

For food items that were insufficiently consumed in a group (e.g., alcoholic drinks in youngest age group) no correlations between the two dietary assessment tools could be calculated. For food items for which there is not enough variation in intakes among the study participants (like for pizza which is highly consumed by all children) the correlations are lower.

## 4. Discussion

The aim of this study was to assess the relative validity against a 3-day diary method in a sample of Italian children and adolescents at least 12 years old up to 17 years old. The overall dietary intake was considered in the present study by assessing the intake of 163 food items related to colorectal cancer in a FFQ.

Compared to the 3-day food diary however, the estimated mean and median intakes as measured by the FFQ are overall higher. Previous studies have reported that FFQs usually overestimate the food and nutrient values compared to other dietary assessment methods [2,10,11] as presented in Table 1, mean and median intakes are overall higher in the FFQs than in the food diaries, except for potatoes, rice, vegetable oils, and butter intake, which were higher in the diaries than in other FFQs. This overestimation might be mainly related to some food items in the FFQ that may not have been consumed during the three days of recall, and also the differences in terms of data collection, questionnaire structure, and time between the surveys. In addition, difficulties in portion size estimations during completion of the 3rd food diary might also bias the true validity of the FFQ. For food groups that are difficult to quantify in standard or household units in the diaries (e.g., pizza or vegetables, meat, potatoes, rice, etc. which are often part of a mixed dish) the difference could be significant between the FFQ and the 3-day diary results. In the 3-day diaries, for food groups like vegetables, meat products, and potatoes, the dietitians coding the diaries had to assign standard portion sizes when the respondent was not able to quantify the consumed amount of food in grams (e.g., during school lunches). Since no standard portion sizes were available for children in Italy, those from the general Italian population had to be used instead for both the FFQ as well as the items with missing portion sizes in the 3-day diaries. It should be noted that these standard portion sizes in the FFQs could have been too high for children and consequently have introduced non-negligible differences between the FFQ and 3-day diary results, particularly for food groups for which respondent portion size information could be used in the 3-day diaries.

The strength of the associations ranged importantly between food groups, going from “no correlation” (e.g., soups) to correlations higher than 0.4 (e.g., coffee and tea). Spearman correlations were highest for coffee and tea (>0.4), followed by alcoholic drinks, soy products, fruits, vegetables, meat products, rice (>0.3). Moderate correlations were found for soft drinks, dairy products, juices, potatoes, breakfast cereals, and sauces. For some food groups, such as dairy, potatoes, and bread, these moderate correlations tend to become higher when stratifying the analyses for age group. Low correlations were found for foods such as eggs, legumes, fish, and culinary ingredients such as vegetable oils and butter consumed more as ingredient or dressing than as main component of the dish. However, all correlations and more in particular those that are less frequently consumed such as eggs and legumes increased importantly when adjusting for within-person variability using the variance ratio.

Previous studies also demonstrated that validity estimates of foods that are frequently consumed have been reported as higher compared to foods that are periodically consumed [12]. A validation study of an FFQ in pre-school aged children in Flanders reported similar correlation coefficients as observed in the present study [13]. A low correlation coefficient was found between the FFQ and 3-day diary for the majority of food groups (<0.30). The largest corrected correlations (>0.6) were found for the intake of potatoes and grains, fruit, milk products, cheese, sugared drinks, and fruit juice, while the lowest correlations (<0.4) for bread and meat products [13]. The present findings are comparable to other studies conducted among child and adolescent populations. Day-to-day variability in children’ diets might also be responsible for these larger differences between the results derived from the FFQ and the 3-day diaries. The low within- over between-individual variability ratio of milk products implies that most children in Italy are consuming milk products on a regular (daily) basis. The high variance ratios for the other food groups however are due to the large day-to-day variability in food consumption. For the food groups showing high variance ratios (e.g., legumes, fish, potatoes, eggs), the reference measurements may be biased and imperfectly reflect ranking. Therefore, Pearson correlations were corrected for attenuation, which improved the correlations for all food groups. Corrected Pearson correlations between the FFQ and the 3d diaries showed values between 0.2 and 0.6. Other validation studies of intake of food groups and single food items assessed by the FFQ have observed correlations, generally between 0.3 and 0.8 [13,14,15,16,17].

Strengths of the present study are the sample size that is above the minimum recommended sample size for dietary validation studies (*n* = 250) of food frequency questionnaires [1,5]. However, there are some limitations that need to be considered when interpreting the results. First, the FFQ contained 163 food items, which is much higher compared to the median number of 88 food items in FFQs [1] and as such potentially imposing a considerable burden on the participants. This high number of foods may also lead to over-reporting of certain food groups [17]. The low number of questionnaires that had to be discarded due to insufficient data indicates satisfactory compliance of the participants. Second, total energy intake was not assessed by the FFQ and could not be used to adjust validity estimates. Third, both the FFQ and diary were self-reported. Although care was taken to provide clear instructions on how to fill out the form, some misreporting cannot be ruled out. In addition, seasonal effects were not assessed. It should also be noted that the 3-day diary has limitations, particularly for estimating usual intakes of foods not consumed on a daily or regular basis such as fish intake and the intake of other specific food groups. In addition, during a focus group meeting with the researchers involved in the fieldwork, it was specified that students had more difficulties in completing the 3-day diaries than in completing the FFQ, which complicated the interpretation of the relative validity results as no golden standard was available. Therefore, the relative validity results should be interpreted with caution as low correlations and large differences may also be due to measurement error in the 3-day diaries for estimating specific foods and food groups.

The authors decided to consider this FFQ as appropriate for use in the bone sarcoma study Metabol Sarc (NCT 04735289). However, it should be noted that in the Metabol Sarc study, the FFQ data will be complemented with levels of dietary metabolites derived from metabolomics analysis to further improve the dietary intake assessment.

## 5. Conclusions

Overall, the relative validity of the Italian EPIC FFQ tailored for use in childhood and/or adolescent populations is satisfactory and could be suitable for ranking individuals according to high and low intakes of most food groups. Nevertheless, it should be noted that large day-to-day variation in food group intakes of the 3d diary data complicated the evaluation of the relative validity of this FFQ. Overall, moderate levels of relative validity were observed for estimates of food group intakes.

Considering the important challenges in assessing dietary intakes among children, this validation study is important as it demonstrates the possibility to use a tailored version of an internationally standardized tool like the EPIC FFQ also for children in large-scale studies.

## Figures and Tables

**Table 1 nutrients-13-01245-t001:** Food groups description of the total study population (mean ± standard deviation; Min, P25, P50, P75 and Max; 95% confidence interval); *n* = 254.

Food Groups	FFQ/Diary	Mean (g)	95% Confidence Interval for Mean		Standard Deviation	Minimum	Maximum	Percentiles		
			Lower Bound	Upper Bound				25	50	75
Alcoholic Drinks	FFQ	22.69	16.25	29.13	52.11	0.00	362.80	0.00	0.00	15.58
	Diary	1.72	0.21	3.24	12.25	0.00	165.00	0.00	0.00	0.00
Soft Drinks	FFQ	115.09	88.91	141.27	211.85	0.00	1600.00	6.70	57.10	114.30
	Diary	70.70	56.51	84.89	114.81	0.00	773.33	0.00	0.00	110.00
Coffee and Tea	FFQ	74.60	64.97	84.24	77.97	0.00	376.00	15.00	51.00	112.83
	Diary	46.60	36.26	56.95	83.70	0.00	520.00	0.00	0.00	64.67
Soup and Bouillon	FFQ	38.00	31.76	44.23	50.46	0.00	350.00	10.70	22.85	44.33
	Diary	0.85	0.26	1.44	4.77	0.00	75.00	0.00	0.00	1.00
Milk and Yoghurt	FFQ	188.38	169.52	207.24	152.63	0.00	934.70	75.13	162.60	252.05
	Diary	130.84	117.62	144.05	106.95	0.00	566.67	28.92	126.17	215.92
Soy Products	FFQ	4.47	2.21	6.73	18.26	0.00	157.70	0.00	0.00	0.40
	Diary	2.34	0.01	4.66	18.84	0.00	166.67	0.00	0.00	0.00
Juices (fruit & vegetable)	FFQ	120.04	102.49	137.59	142.06	0.00	1125.00	29.20	71.40	160.70
	Diary	48.52	38.88	58.16	77.99	0.00	333.33	0.00	1.50	71.92
Fruit	FFQ	264.96	244.71	285.22	163.90	0.00	955.60	155.83	236.50	357.98
	Diary	79.08	67.94	90.22	90.16	0.00	414.33	0.00	53.50	118.33
Vegetables	FFQ	166.44	153.37	179.50	105.72	7.40	730.20	86.70	147.35	218.30
	Diary	164.44	151.23	177.65	106.89	2.33	622.67	89.33	140.50	220.67
Legumes	FFQ	20.18	17.97	22.39	17.86	0.00	93.70	6.95	15.85	29.68
	Diary	10.79	8.06	13.52	22.10	0.00	143.67	0.00	0.00	13.67
Potatoes	FFQ	40.94	37.19	44.69	30.35	0.00	179.60	21.63	31.20	48.30
	Diary	68.76	60.24	77.29	69.01	0.00	294.00	0.00	59.17	103.42
Meat	FFQ	192.55	180.07	205.03	101.00	1.60	686.40	112.45	184.05	245.73
	Diary	139.36	129.42	149.30	80.44	10.00	475.00	85.17	120.00	178.00
Eggs	FFQ	26.65	24.38	28.93	18.44	1.10	105.20	14.30	22.50	32.70
	Diary	24.37	21.28	27.46	25.02	0.00	109.33	5.00	16.83	37.67
Fish	FFQ	39.79	36.48	43.09	26.75	0.00	150.20	20.65	36.25	57.00
	Diary	35.38	29.04	41.71	51.27	0.00	281.33	0.00	12.83	56.67
Biscuits and Candies	FFQ	135.35	123.26	147.44	97.86	5.40	655.50	69.28	110.65	173.05
	Diary	97.53	43.23	151.84	439.50	0.00	7011.25	27.25	54.17	103.33
Breakfast Cereals	FFQ	7.25	5.66	8.84	12.85	0.00	80.00	0.00	0.40	9.30
	Diary	4.01	2.44	5.58	12.69	0.00	121.67	0.00	0.00	0.00
Bread	FFQ	77.02	69.73	84.30	58.95	0.00	319.80	34.15	62.05	102.55
	Diary	62.75	55.94	69.55	55.06	0.00	263.33	19.58	50.00	97.42
Rice	FFQ	10.54	8.83	12.24	13.80	0.00	96.00	2.40	5.40	13.98
	Diary	20.75	15.65	25.86	41.31	0.00	333.33	0.00	0.00	29.00
Grains. including Pizza	FFQ	132.16	124.33	139.99	63.35	19.60	398.30	89.30	123.10	161.13
	Diary	119.81	111.87	127.76	64.28	0.00	456.33	78.25	109.17	146.42
Vegetable Oil	FFQ	22.58	20.82	24.33	14.19	0.00	85.00	12.53	20.50	30.55
	Diary	33.30	31.37	35.23	15.63	8.00	134.33	21.67	31.33	42.00
Butter	FFQ	2.25	1.77	2.73	3.88	0.00	26.10	0.20	0.70	2.83
	Diary	3.47	2.70	4.25	6.27	0.00	52.00	0.00	0.00	4.67
Sauces	FFQ	2.88	2.45	3.30	3.43	0.00	20.00	0.38	1.40	4.00
	Diary	1.51	1.03	1.98	3.88	0.00	25.00	0.00	0.00	0.00
Sugar. Honey. Jam	FFQ	15.73	13.32	18.13	19.47	0.00	129.70	4.15	9.30	20.30
	Diary	5.83	4.60	7.06	9.95	0.00	72.00	0.00	0.67	8.08

**Table 2 nutrients-13-01245-t002:** Spearman correlations corrected for within person variability of 3-day diaries; *n* = 254.

Food Groups	Correlation for				
	Overall Population	Women	Men	Youngest Age Group Low Grade School (12–13 years)	Oldest Age Group High Grade School (14–17 years)
Alcoholic Drinks	0.357	--	0.444	--	0.353
Soft Drinks	0.352	0.289	0.404	0.301	0.410
Coffee and Tea	0.509	0.491	0.522	0.524	0.445
Soup and Bouillon	--	--	--	--	--
Milk and Yoghurt	0.395	0.378	0.308	0.365	0.445
Soy Products	0.207	0.220	0.215	--	0.262
Juices (fruit & vegetable)	0.361	0.276	0.429	0.304	0.447
Fruit	0.439	0.389	0.475	0.399	0.493
Vegetables	0.517	0.467	0.544	0.453	0.608
Legumes	0.587	0.466	0.695	0.746	--
Potatoes	0.470	0.556	--	--	0.716
Meat	0.459	0.407	0.454	--	0.582
Eggs	--	--	0.447	--	0.577
Fish	--	0.655	--	--	0.668
Biscuits and Candies	0.407	0.414	0.396	--	0.617
Breakfast Cereals	0.415	0.385	0.432	0.365	0.478
Bread	0.276	--	0.285	--	0.486
Rice	0.347	0.397	0.323	0.346	--
Grains, including Pizza	0.357	0.355	0.314	--	0.397
Vegetable Oil	0.214	--	0.333	--	0.272
Butter	0.439	0.477	--	--	0.518
Sauces	0.350	--	0.541	--	0.526
Sugar, Honey, Jam	0.283	--	0.354	0.308	0.269

## Data Availability

Access to data can be requested to the corresponding author.

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
