# Peer review of "Relative Validity of an Italian EPIC Food Frequency Questionnaire for Dietary Factors in Children and Adolescents. A Rizzoli Orthopedic Institute Study"

_nutrients, 2021, doi:10.3390/nu13041245_

Round 1
Reviewer 1 Report
It has been really a pleasure to read this manuscript. It was very necessary to perform this validation to help better understanding of lifestyle factors influencing cancer at early stages. I only have to add some minor corrections to this very clear and nice report.
Abstract: Aimed to evaluate?
Intro:
Line 57: which factors? It seems the sentence is not finished…
Line 152: there is a typo error: divided?
Methods:
A bit of context on the pictures of the recipes or food groups will increase the understanding and the interpretation of the results.
Results:
There is a lack of context why the analyses are stratified by age based on the age of 15y…
You did not explain in the methodology how did you calculate the within person variability…
Discussion:
Typo error in 322: dietitians
Lines 342-344: I would not say that eggs, butter or vegetable oils are not regularly consumed food groups. Another explanation for them?
Due to the relatively small correlations found, do you plan to make any potential correction or measure to increase the validity or the congruence with the reality? This might require a bit of discussion.
Author Response
Dear Reviwer,
we made corrections according to your suggestions and point by point answers are uploaded
best regards
Alessandra Longhi

Reviewer 2 Report
This was a well powered study to validate a FFQ against a 3 day diary in children aged 12-17 y. There are too few studies of this type.
Table 1 Are values daily?
Author Response
Dear Reviwer
thank you for your comment
WE answer in the tables: food is expressed in grams
Best regards
Reviewer 3 Report
Huybrechts et al. submitted to Nutients as a research article: “Relative validity of a food frequency questionnaire for dietary factors in children and adolescents to be used in a bone sarcoma study”.
The manuscript by Dr. Huybrechts et al. includes 254 healthy adolescents from several High and Middle schools in Italy. They received an adapted version of the EPIC FFQ and filled in a 3-day food diary for validation.
Although I highly support extending the current knowledge on dietary intake in childhood cancer patients and survivors, this study is conducted in a healthy population. Therefore, the adapted EPIC FFQ can be used for a broader population than only for soft tissue sarcoma patients. The link with bone sarcomas is irrelevant for this manuscript and should be removed.

Author Response
Dear Reviewer
thank you for your suggestions
we made changes according to your queries
The title, abstract and introduction were rephrased. see attached point by point answer
best regards

Round 2
Reviewer 3 Report
Please, see the attached PDF file.